# FGW-CLIP: ENHANCING ENZYME SCREENING VIA FUSED GROMOV-WASSERSTEIN CONTRASTIVE LEARNING

## ABSTRACT

Enzymes are crucial catalysts for biochemical reactions, underpinning numerous biological processes. The efficient identification of specific enzymes from extensive protein libraries is essential for understanding and harnessing these biological reactions. While traditional computational methods for enzyme screening are time-consuming and resource-intensive, recent contrastive learning approaches have shown promise. However, these methods often overlook the inherent hierarchical classifications within enzymes and reactions, as well as the significance of molecular structure in catalysis. To address these limitations, we introduce FGW-CLIP, a novel contrastive learning framework based on optimizing the fused Gromov-Wasserstein distance. This approach incorporates multiple alignments, including representation alignment between reactions and enzymes, and internal alignment within enzyme and reaction representations. By introducing a regularization term, our method minimizes the Gromov-Wasserstein distance between enzyme and reaction spaces, enhancing information exchange within these domains. FGW-CLIP demonstrates superior performance on the widely-used EnzymeMap benchmark, significantly outperforming existing methods in enzyme virtual screening tasks. Notably, it achieves state-of-the-art results in both BEDROC and EF metrics, indicating its efficacy in identifying relevant enzymes for given reactions. These results highlight the potential of our method to advance virtual enzyme screening, offering a powerful tool for enzyme discovery and characterization.

## 1 INTRODUCTION

Enzymes play a vital role in various biological processes such as biosynthesis. They act as catalysts, speeding up chemical reactions in living organisms. However, in the vast protein sequence databases, such as UniProt, only about 1/5 of proteins have been experimentally verified, and only 0.23% have received sufficient attention from researchers(Ribeiro et al., 2023). Effective enzymes may be in the billions of unexplored sequences.

Traditional calculation methods, including sequnce-similarity–based (Altschul et al., 1990; Desai et al., 2011; Altschul et al., 1997), homology-based (Krogh et al., 1994; Steinegger et al., 2019), structure-based (Roy et al., 2012; Zhang et al., 2017) methods, consume a large amount of manpower and material resources. And they are faced with protein annotation errors in calculation methods. In recent years, machine learning based methods have emerged, primarily utilizing contrastive learning techniques on the screening of enzymes. CLEAN (Yu et al., 2023b) improved EC number assignment to enzymes, annotating understudied ones accurately and correcting mislabeled entries. While CLIPZyme (Mikhael et al., 2024) is a computational framework that encodes and aligns enzyme structure and reaction pair representations for in-silico enzyme screening. However, they focus on enzyme-reaction relationship but overlook inherent hierarchical classifications within them and the importance of molecular structure in catalysis.

In this work, we introduce FGW-CLIP, a novel contrastive learning framework based on the optimization of fused Gromov-Wasserstein distance. This framework incorporates multiple alignment methods, including representation alignment between reactions and enzymes (similar to CLIP), as well as internal alignment of enzyme and reaction representations. By introducing regularization

terms, our method minimizes the Gromov-Wasserstein distance between the enzyme and reaction spaces during model training, thereby enhancing information interaction in both domains.

We provide theoretical insights into FGW-CLIP from the perspective of optimizing the fused Gromov-Wasserstein distance. Additionally, we offer empirical support by validating our approach on the widely-used EnzymeMap benchmark for enzyme virtual screening. FGW-CLIP outperforms existing baselines, achieving state-of-the-art results on Boltzmann-enhanced discrimination of ROC (BEDROC) and enrichment factor (EF) metrics.

Our key contributions are as follows:

- We propose a novel framework for enhancing contrastive learning by optimizing the fused Gromov-Wasserstein distance. This approach considers not only the alignment of reactions and enzymes but also the internal alignment within each domain. Furthermore, we introduce a regularization term based on the Gromov-Wasserstein distance to preserve structural information within individual spaces while maximizing the alignment between reaction and enzyme spaces.
- We provide theoretical insights into the proposed contrastive learning framework, exploring the foundations of using fused Gromov-Wasserstein distance in the context of enzyme-reaction alignment.
- Our framework achieves state-of-the-art results on the widely used EnzymeMap benchmark dataset. We present detailed ablation studies to demonstrate the importance of incorporating internal structural information and Gromov-Wasserstein loss in our approach.

## 2 RELATED WORK

### 2.1 CONTRASTIVE LEARNING

Contrastive learning has found extensive applications in vision and multimodal representation learning. CLIP (Contrastive Language-Image Pretraining) enhances multimodal contrastive learning by effectively combining image and text information, making it widely applicable in fields such as image classification, text generation, and human-computer interaction. MLIP (Zhang et al., 2024) enhances CLIP by integrating spatial and frequency-domain information, improving multimodal learning through a multi-perspective approach. iCLIP (Wei et al., 2023) bridges the gap between image classification and contrastive learning, optimizing CLIP for both visual tasks and language-image pairings. X-MoRe (Eom et al., 2023) refines CLIP's embeddings to enhance performance in image-to-text and text-to-image retrieval tasks, improving its adaptability for real-world applications.

### 2.2 FUSED GROMOV-WASSERSTEIN DISTANCE

The Gromov-Wasserstein (GW) Distance is a metric used in optimal transport theory that measures the similarity between two metric spaces by considering the structures of the spaces rather than their individual points. Mémoli (Scetbon et al., 2022) proves that $GW^{1/2}$ defines a distance on the space of metric measure spaces quotiented by measure-preserving isometries. Fused Gromov-Wasserstein (FGW) (Titouan et al., 2019; Ma et al., 2024)distance extends the Gromov-Wasserstein metric to calculate transportation distance between two unregistered probability distributions on different product metric spaces, such as combining graph signals and structures, making it suitable for attributed graphs.

### 2.3 ENZYME SCREENING

Enzyme virtual screening and recognition accelerate the discovery of new enzymes and drug candidates by accurately identifying functions and efficiently screening potential inhibitors from large libraries. CLIPZyme (Mikhael et al., 2024) serves as a computational framework and effectively encode and align representations of enzyme structures and their corresponding reaction pairs for in-silico enzyme screening. CLEAN (Yu et al., 2023b) improved the assignment of EC numbers to enzymes, accurately annotating understudied enzymes and correcting mislabeled entries . Moreover, sequence similarity-based tools or ML models such as BLASTp (Altschul et al., 1990), DeepEC (Wang et al., 2020), and ProteInfer (Sanderson et al., 2023), can also be used to predict EC numbers.

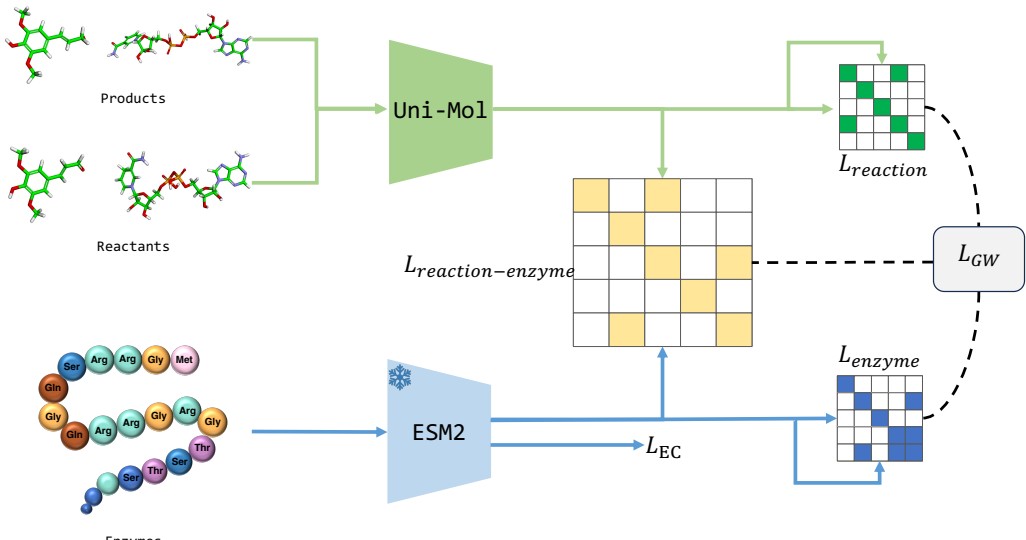

Figure 1: Overview of FGW-CLIP Framework

# 3 METHOD

## 3.1 OVERVIEW OF FGW-CLIP

Giving huge protein libraries, enzyme screening is to identify enzymes that can catalyze specific chemical reactions from the libraries. We consider this task as a dense retrieval issue. Trained encoders produce representations for both reactions and enzymes. Subsequently, reactions are used as queries, and enzymes are ranked according to their cosine similarity to these reactions. The top enzymes having the highest similarity are then recognized as the most probable candidates for catalyzing the given reaction.

We propose FGW-CLIP from the perspective of optimizing the fused Gromov-Wasserstein distance. As shown in Figure 1, we use the pretrained 3D molecular model, Uni-Mol Zhou et al. (2023), to encode the molecules in a reaction, and obtain the reaction embedding through a readout function. For enzymes, we employ the pretrained protein language model, ESM2 Lin et al. (2023), to derive enzyme sequence embeddings. During subsequent training of FGW-CLIP, the ESM2 model remains frozen. We perform contrastive learning between reactions and enzymes, where reactions that can be catalyzed by a given enzyme serve as positive samples, while others serve as negative samples. Additionally, we capture the intrinsic connections within enzymes and reactions: based on the Enzyme Commission (EC) number. Specifically, each data point has an EC number. Since enzymes or reactions can have multiple EC numbers, we add a list of EC numbers to each data point. Taking reactions as an example, if the original EC number of a reaction is present in the EC number list of another data point in the batch, we consider them a positive pair; if it is not in the list, they form a negative pair. To better leverage the structural information inherent in both the enzyme and reaction spaces, and to ensure consistency between these structures, we introduce a novel regularization term based on the optimization of the Gromov-Wasserstein distance. Furthermore, we incorporated a loss function for predicting the EC number of enzymes into the FGW-CLIP framework. We found that this framework can be easily extended for efficient prediction of enzyme EC numbers.

In the following sections, we provide a more detailed explanation of the core components and training strategies. In Section 3.2, we introduce the molecular encoder, Uni-Mol and the enzyme encoder, ESM2. In Section 3.3, we delve into the training strategy of FGW-CLIP for contrastive learning between reactions and enzymes. In Section 3.4, we analyze FGW-CLIP from the perspective of optimizing the fused Gromov-Wasserstein distance, offering insights into its methodological advantages.

## 3.2 PRETRAINING BACKBONE OF REACTION AND ENZYME

For the reaction representation, Uni-Mol is utilized to encode the molecules, and the reaction embedding is obtained via a readout function. Uni-Mol integrates 3D structural information during encoding, which is crucial in the catalytic process, as some enzymes interact with intermediate products to exert their catalytic function. Uni-Mol is pretrained on large-scale molecules and their conformations. It leverages distance-based attention bias to integrate the 3D information of molecules into the encoding. This 3D incorporation allows Uni-Mol to excel in tasks like molecular property prediction and protein-ligand binding pose prediction. We obtain the embedding of the entire molecule using a CLS token and normalize it with Euclidean norm.

For enzyme encoding, we use ESM2, a protein language model pretrained on millions of protein sequences. One of its key strengths is its ability to handle longer sequences and generate detailed embeddings, making it highly suitable for various protein-related tasks, such as structure and function prediction. In our framework, we leverage the pretrained ESM2 model to encode enzyme sequences, using the average of all residue embeddings to represent the entire protein, and it remains frozen during subsequent training.

## 3.3 TRAINING STRATEGY OF FGW-CLIP

### 3.3.1 CONTRASTIVE LEARNING STRATEGY

In this section, we introduce the contrastive learning strategy used in FGW-CLIP. We utilize the reaction-enzyme catalysis data from EnzymeMap, where each data entry consists of a reaction, the enzyme that catalyzes it, and an EC number. We perform contrastive learning between reactions and enzymes.

Since the relationship between reactions and enzymes is not strictly one-to-one, we construct a ground truth label matrix from the data. In this matrix, the rows represent reactions and the columns represent enzymes. If a reaction can be catalyzed by a specific enzyme, the corresponding position in the matrix is set to 1; otherwise, it is set to 0. We set the index dataset $E_i, R_i$ of reaction $i$ and enzyme $i$. The loss function is formulated as follows, following the InfoNCE loss:

$$L_{\text{reaction-enzyme}} = \frac{1}{2} \sum_{i=1}^{N} \left[ - \sum_{j \in R_i} \log \frac{e^{(\text{sim}(r_i, e_j)/\tau)}}{\sum_{k=1}^{N} e^{(\text{sim}(r_i, e_k)/\tau)}} - \sum_{j \in E_i} \log \frac{e^{(\text{sim}(e_i, r_j)/\tau)}}{\sum_{k=1}^{N} e^{(\text{sim}(e_i, r_k)/\tau)}} \right] \quad (1)$$

In addition to modeling the catalytic relationship between reactions and enzymes, we also consider the internal relationships within enzymes and reactions. We set the index dataset $I_i$ of item $i$. The InfoNCE loss for this internal relationship modeling is given as:

$$L_{\text{internal}} = - \sum_{i=1}^{N} \sum_{j \in I_i} \log \frac{e^{(\text{sim}(x_i, x_j)/\tau)}}{\sum_{k=1}^{N} e^{(\text{sim}(x_i, x_k)/\tau)}} \quad (2)$$

Here, $x_i$ and $x_j$ represent the embeddings of either enzymes or reactions that share the same EC number, while $x_k$ includes all possible embeddings in the same batch.

### 3.3.2 EC PREDICTION

The Enzyme Commission (EC) number is a standardized numerical classification scheme for enzymes based on the chemical reactions they catalyze. Each EC number consists of four hierarchical levels that describe the enzyme's function in increasing specificity. By identifying the EC numbers, we can narrow down the possible enzymes that could catalyze specific reactions, thereby improving the efficiency and accuracy of enzyme screening.

In our approach, we utilize the enzyme's representation to predict the EC classes at all four hierarchical levels. For each level, a separate classification head is employed, and we use cross-entropy loss for the predictions. The loss for a single level $i$ is defined as:

$$L_{\text{level } i} = -\sum_{c=1}^{C_i} y_c \log(p_c) \tag{3}$$

Here, $C_i$ represents the number of classes at level $i$, $y_c$ is the true label (one-hot encoded), and $p_c$ is the predicted probability for class $c$.

The total EC classification loss is the sum of the losses for the four levels:

$$L_{\text{EC}} = L_{\text{level } 1} + L_{\text{level } 2} + L_{\text{level } 3} + L_{\text{level } 4} \tag{4}$$

This multi-level classification allows us to capture the hierarchical nature of enzyme functions. By incorporating the EC classification head, we not only improve enzyme screening but also ensure that the learned enzyme embeddings contain rich functional information.

### 3.3.3 REGULARIZATION LOSS FOR GW DISTANCE OPTIMIZATION

To advance the alignment between the reaction space and the enzyme space, we introduce an additional regularization term motivated by the goal of minimizing the Gromov-Wasserstein (GW) distance.

This regularization is essential to maintain the internal structure of both spaces while aligning them effectively. The complete loss function is given as:

$$L_{\text{GW}} = -\sum_{i,j,i',j'=1}^{N} \text{sim}(e_i, r_j) \cdot \text{sim}_d(r_j, r_{j'}) \cdot \text{sim}(r_{i'}, e_{j'}) \cdot \text{sim}_d(e_i, e_{i'}) \tag{5}$$

Here, $\text{sim}(e_i, r_j)$ represents the similarity between enzyme ($e_i$) and reaction ($r_j$) embeddings. $\text{sim}_d(r_j, r'_j)$ and $\text{sim}_d(e_i, e'_i)$ denote the internal similarities within the reaction and enzyme spaces, respectively. These internal similarities are detached during training, meaning their gradients are not propagated to avoid interfering with the optimization of other terms. This objective encourages alignment between the reaction and enzyme spaces using the internal structural information of each space.

### 3.4 FGW-CLIP: ENHANCING CLIP BY OPTIMIZING GROMOV-WASSERSTEIN DISTANCE

By integrating the training objectives in Section 3.3, we can derive the overall training objective for FGW-CLIP, denoted as $L_{\text{FGW}}$, as follows:

$$L_{\text{FGW}} = (1 - \alpha)(L_{\text{reaction-enzyme}} + L_{\text{reaction}} + L_{\text{enzyme}}) - 2\alpha L_{\text{GW}} + \lambda L_{\text{EC}}$$

The loss function involves multiple terms. Based on the approach outlined in Shi et al. (2023)Zhou et al. (2024), we establish a connection between $L_{FGW}$ and the fused Gromov-Wasserstein distance optimization problem under a specific constraint through the proposition 1.

**Proposition 1** *Given encoder $f_{\psi_1}$ for data field $X_1$ and encoder $f_{\psi_2}$ for data field $X_2$, $x_{\psi_1}$ represents the l2 normalized embeddings of $X_1$ from $f_{\psi_1}$, while $x_{\psi_2}$ represents the l2 normalized embeddings of $X_2$ from $f_{\psi_2}$. $\Gamma^{f_1}$ represents the label on $X_1$, $\Gamma^{f_2}$ represents the label on $X_2$, $\Gamma^{cor}$ represents the label on the pairs $(X_1, X_2)$. FGW-CLIP could be derived from optimizing a specific constraint-fused Gromov-Wasserstein distance as follows:*

$$\min_{\theta, \psi_1, \psi_2} \left\{ (1 - \alpha) KL(\Gamma^{cor} || \Gamma^{\theta}) + \alpha GW(\Gamma_d^{\psi_1}, \Gamma_d^{\psi_2}, \Gamma^{\theta}) \right.$$

$$+ \lambda_1 KL(\Gamma^{f_1} || \Gamma^{\psi_1}) + \lambda_2 KL(\Gamma^{f_2} || \Gamma^{\psi_2}) - \lambda_{ce} CE(y_{\psi_1}, f_{\psi_1}(X_1)) \bigg\}$$

$$\text{subject to} \quad \Gamma^{\theta} = \arg\min_{\Gamma \in U(a^{cor})} \left( \langle C^{\theta}, \Gamma \rangle - \tau H(\Gamma) \right),$$

$$\Gamma^{\psi_1} = \arg\min_{\Gamma \in U(a^{\psi_1})} \left( \langle C^{\psi_1}, \Gamma \rangle - \tau H(\Gamma) \right),$$

$$\Gamma^{\psi_2} = \arg\min_{\Gamma \in U(a^{\psi_2})} \left( \langle C^{\psi_2}, \Gamma \rangle - \tau H(\Gamma) \right),$$

(6)

where $KL(X||Y) = \sum_{ij} x_{ij} log \frac{x_{ij}}{y_{ij}} - x_{ij} + y_{ij}$ represents the Kullback-Leibler divergence, and $H(\Gamma) = -\sum_{i,j} \Gamma_{ij}(\log(\Gamma_{ij}) - 1)$ represents entropic regularization. $\Gamma^\theta, \Gamma^{\psi_1}, \Gamma^{\psi_2} \in R_+^{N \times N}$, $C^\theta, C^{\psi_1}, C^{\psi_2} \in R_+^{N \times N}$ are cost matrix and $C^\theta(i,j) = c - x_{\psi_1,i} x_{\psi_2,j}^T$, $C^{\psi_1}(i,j) = c - x_{\psi_1,i} x_{\psi_1,j}^T$, $C^{\psi_2}(i,j) = c - x_{\psi_2,i} x_{\psi_2,j}^T$. $a^{\psi_1}, a^{\psi_2}, a^{cor}$ represent the label vector of dataset $X_1$, $X_2$ and pair dataset $(X_1, X_2)$. $\Gamma_d^{\psi_1}, \Gamma_d^{\psi_2}$ represent the values of $\Gamma^{\psi_1}, \Gamma^{\psi_2}$ respectively, with the gradients detached, $GW(\Gamma_d^{\psi_1}, \Gamma_d^{\psi_2}, \Gamma^\theta) = \sum_{i,j=1}^n \sum_{i',j'=1}^n |\Gamma_d^{\psi_1}(i,i') - \Gamma_d^{\psi_2}(j,j')|^2 \Gamma^\theta(i,j) \Gamma^\theta(i',j')$ is the Gromov-Wasserstein distance. $CE$ is the cross-entropy loss of data field $X_1$, which is added to facilitate a specific classification task as a regularization term.

The proof is provided in the Appendix A.2. In $L_{FGW}$, we utilize $\Gamma_{\psi_1}$ and $\Gamma_{\psi_2}$ to learn the structural information of two data domains $X_1$ and $X_2$, respectively. Through the optimization of the Gromov-Wasserstein distance, structural alignment at the domain level is achieved. We consider this overall structural alignment information as a supplement and enhancement to the existing label alignment information between the two domains $X_1$ and $X_2$. By optimizing this fused Gromov-Wasserstein distance, we can better extend the generalization capability of the CLIP model and alleviate the issue of insufficient effective labels between domains $X_1$ and $X_2$.

# 4 EXPERIMENT

## 4.1 ENZYME VIRTUAL SCREENING

### 4.1.1 DATASETS

**EnzymeMap** Based on the original EnzymeMap dataset (Heid et al., 2023), it involves biochemical reactions linked to UniProt IDs and EC numbers. There are 46,356 enzyme-driven reactions with 16,776 unique chemical reactions, 12,749 enzymes, 2,841 EC numbers, and 394 reaction rules in the EnzymeMap dataset. We split the dataset into training, validation, and test sets based on the reaction rule IDs, with a ratio of 0.8/0.1/0.1, containing 34,427, 7,287, and 4,642 entries, respectively, the same as in CLIPZyme.

**Enzyme Screening Set** This dataset integrated the EnzymeMap dataset, Brenda release 2022_2 (Chang et al., 2020), and UniProt release 2022_01 (Yu et al., 2023a), and filtered out the sequences that are longer than 650 amino acids. It includes a total of 261,907 protein sequences. Enzyme screening Set is used as a virtual screening database, where we use the reactions from the EnzymeMap test set as queries to perform screening in it.

### 4.1.2 BASELINE

In this task, we use the state-of-the-art method CLIPZyme(Mikhael et al., 2024) as the baseline, which is a contrastive learning approach for enzyme screening. We follow the experimental setup of CLIPZyme and use the same datasets.

### 4.1.3 EVALUATION METRIC

For this task, we utilize the BEDROC (Boltzmann-Enhanced Discrimination of ROC) (Truchon & Bayly, 2007) score and the enrichment factor (EF) as evaluation metrics. We calculate BEDROC at $\alpha = 85$ and $\alpha = 20$, and focus on EF in the top 5% and 10% of the predictions, to align with the evaluation protocol in CLIPZyme.

### 4.1.4 RESULTS

Table 1 shows the performance comparison between FGW-CLIP and the current SOTA baseline CLIPZyme on EnzymeMap, with the best results highlighted in bold. We also compared the results of CLIPZyme using different protein and reaction encoders. CGR(Hoonakker et al., 2011) is a method for obtaining reaction representations based on graph structures. ESM indicates that ESM is used for finetuning to obtain enzyme representations. CGR The results for CLIPZyme are consistent

Table 1: Enzyme virtual screening performance on EnzymeMap. The higher the BEDROC and EF, the better.

| Method | BEDROC$_{85}$(%) | BEDROC$_{20}$(%) | EF$_{0.05}$ | EF$_{0.1}$ |
|---|---|---|---|---|
| CLIPZyme (ESM) | 36.91 | 53.04 | 11.93 | 6.84 |
| CLIPZyme (CGR) | 38.91 | 57.58 | 13.16 | 7.73 |
| CLIPZyme | 44.69 | 62.98 | 14.09 | 8.06 |
| FGW-CLIP | **48.66** | **66.69** | **14.91** | **8.18** |

with those reported in their original paper. As shown in the table, FGW-CLIP achieves the best performance across all four metrics for BEDROC and EF, with significant improvements of about 4% on both BEDROC$_{85}$ and BEDROC$_{20}$. This demonstrates the advantage of FGW-CLIP in optimizing the fused GW distance through contrastive learning, focusing on both the alignment between reactions and enzymes and the internal relationships within enzymes and reactions.

Table 2: Ablation studies performance on EnzymeMap. Exclude enzymes that appeared in the training set from the screening set.

| Exclusion Criteria | Method | BEDROC$_{85}$(%) | BEDROC$_{20}$(%) | EF$_{0.05}$ | EF$_{0.1}$ |
|---|---|---|---|---|---|
| Exact Match | Clipzyme | 39.13 | 58.86 | 13.40 | **7.81** |
| | FGW-CLIP | **45.14** | **61.43** | **13.57** | 7.61 |

We also evaluate the generalization capabilities of FGW-CLIP and the baseline CLIPZyme. Specifically, we conduct an experiment focusing on unseen enzymes. We exclude any enzymes in the screening set that appeared in the training set. Table 2 presents the results, showing that FGW-CLIP outperforms CLIPZyme on 3 out of the 4 evaluation metrics, and is nearly close on EF$_{0.1}$. Notably, FGW-CLIP achieves a significant lead in BEDROC, indicating its ability to identify relevant enzymes in the absence of prior exposure to them. This indicates the strength of FGW-CLIP in capturing the essential features of enzyme-reaction interactions.

## 4.2 ABLATION STUDY

Table 3: Ablation study of different training strategies on FGW-CLIP's performance on EnzymeMap. "R" represents the reaction, "E" represents the enzyme, and "_" indicates the use of contrastive learning between both sides. "EC" represents the addition of an EC prediction head.

| Method | BEDROC$_{85}$(%) | BEDROC$_{20}$(%) | EF$_{0.05}$ | EF$_{0.1}$ |
|---|---|---|---|---|
| R_E | 45.94 | 61.11 | 13.28 | 7.41 |
| R_E + R_R | 48.08 | 63.92 | 13.89 | 7.89 |
| R_E + EC | 45.25 | 63.93 | 14.46 | 7.97 |
| R_E + R_R + E_E + EC | 45.83 | 64.17 | 14.37 | 8.12 |
| FGW-CLIP | **48.66** | **66.69** | **14.91** | **8.18** |

We conducted comprehensive ablation studies to evaluate the components of FGW-CLIP. First, we evaluate the impact of different training strategies. From R_E and R_E + R_R, it can be observed that removing R_R significantly affects the BEDROC, indicating that capturing the internal relationships between enzymes is crucial for enzyme screening. Comparing R_E to FGW-CLIP, removing EC impacts the EF, suggesting that predicting EC numbers helps the model better focus on relevant enzyme properties. From R_E + R_R + E_E + EC to FGW-CLIP, it can be seen that the complete FGW-CLIP framework brings significant improvements. This addition further elevated the performance across all metrics, underscoring the importance of optimizing the Gromov-Wasserstein distance in aligning the reaction and enzyme spaces effectively.

In addition, we also explore the impact of different $\alpha$ weights and various approaches to incorporating the GW loss into the FGW-CLIP framework. Table 4 presents the results of these ablation studies.

Table 4: Ablation studies performance on EnzymeMap.

| Method | BEDROC$_{85}$(%) | BEDROC$_{20}$(%) | EF$_{0.05}$ | EF$_{0.1}$ |
|---|---|---|---|---|
| $\alpha = 0.05$ | 45.59 | 63.84 | 14.21 | **8.24** |
| $\alpha = 0.3$ | 47.46 | 64.86 | 14.29 | 8.02 |
| $\alpha = 0.5$ | 47.44 | 64.31 | 14.18 | 7.89 |
| Label, $\alpha = 0.1$ | 47.08 | 65.28 | 14.64 | 8.08 |
| No detach, $\alpha = 0.1$ | 47.63 | 64.96 | 14.39 | 8.06 |
| Detach, $\alpha = 0.1$ (FGW-CLIP) | **48.66** | **66.69** | **14.91** | 8.18 |

First, we experimented with different $\alpha$ weights for the GW loss. We found that an $\alpha$ value of 0.1 yielded the best performance. When $\alpha = 0.05$, the effect of the GW loss was minimal, indicating that the influence of the GW loss on aligning the reaction-enzyme space was too weak. Conversely, when $\alpha = 0.5$, the performance decreased, suggesting that a high $\alpha$ disrupted the alignment of the reaction-enzyme space and hindered the learning of the internal structures of enzymes and reactions. Next, we investigated different ways of incorporating the GW loss into the FGW-CLIP framework. The method labeled as 'Label' involves using the labels from the internal contrastive learning (as defined in Eq.2) to replace the similarity matrices $\text{sim}_d(r_j, r_{j'})$ and $\text{sim}_d(e_i, e_{i'})$ in Eq.5. The method labeled as 'No detach' refers to not detaching the similarity matrices $\text{sim}_d(r_j, r_{j'})$ and $\text{sim}_d(e_i, e_{i'})$ in Eq.5, allowing them to participate in gradient backpropagation and adding an optimization term for these matrices.

As shown in the table, our current approach of detaching the similarity matrices provides the best results, making it the optimal choice for building the FGW-CLIP framework. This method balances the alignment of the reaction-enzyme spaces while preserving the internal structures, confirming the effectiveness of our design.

Furthermore, to visually demonstrate the distinctions between embeddings learned by FGW-CLIP and those from pretrained ESM2 checkpoint, we present a comparative visualization in Figure 2. The enzymes depicted are sourced from EnzymeMap, with distinct colors representing different top-level EC numbers ranging from EC 1 to EC 6. Upon comparison, the classification boundaries in Figure 2 generated by FGW-CLIP exhibit greater clarity, and the intra-class molecular distances appear more appropriately scaled. Notably, some clusters subdivide into multiple subclusters, potentially reflecting the inherent hierarchical structure within the molecular compositions.

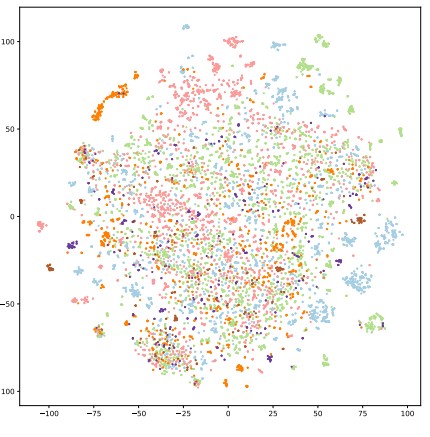
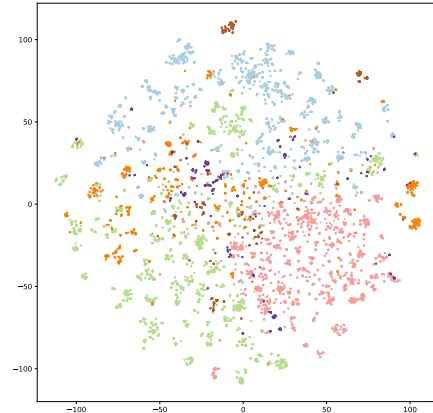

(a) Representations from pretrained checkpoint      (b) Representations learned by FGW-CLIP

Figure 2: t-SNE visualization of enzyme representations learned by pretrained checkpoint versus FGW-CLIP. Different colors represent different top-level EC numbers ranging from EC 1 to EC 6

## 5 CONCLUSION

In this work, we proposed FGW-CLIP, a novel contrastive learning framework that enhances enzyme screening through fused Gromov-Wasserstein distance (FGW) optimization. Our method optimizes fused Gromov-Wasserstein distance to align the reaction and enzyme spaces more effectively while preserving the internal structures of both. We conduct contrastive learning between reactions and enzymes, enzymes and enzymes, along with reactions and reactions. We also introduce an auxiliary loss to predict EC number. Finally, we add the GW loss to form the complete FGW-CLIP framework. In this framework, the model can effectively capture the intricate relationships between enzymes and reactions, leading to a more accurate and robust outcome. We conduct extensive experiments on the EnzymeMap dataset, where FGW-CLIP demonstrated its superiority over the state-of-the-art baseline. Notably, our method achieved significant improvements in key evaluation metrics such as BEDROC and EF, indicating its strong generalization capability and effectiveness in enzyme screening tasks.

In the future, we aim to incorporate enzyme structures into the model and explore further optimization of enzyme functions. Additionally, we plan to investigate the application of FGW-CLIP in other biochemical tasks.

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

# A APPENDIX

## A.1 EVALUATION METRICS

Here we introduce the two metrics we use to evaluate the efficiency of the enzyme virtual screening task. The BEDROC score is a modified version of the AUC of the ROC curve, which places a stronger emphasis on early enrichment (i.e., at high-ranking positions). This is particularly important in drug discovery, where experimental testing is costly, and being able to identify potentially active compounds early on can save significant time and resources. The calculation formula of BEDROC is shown as follows :

$$BEDROC = \frac{\sum_{i=1}^{n} e^{-\alpha r_i/N}}{\frac{R_a(1-e^{-\alpha})}{e^{\alpha/N}-1}} \times \frac{R_a \sinh(\alpha/2)}{\cosh(\alpha/2) - \cosh(\alpha/2 - \alpha R_a)} + \frac{1}{1 - e^{\alpha(1-R_a)}}$$

where $n$ is the number of active compounds; $N$ is the total number of compounds; $R_{=n/N}$ is the ratio of the number of active compounds to the total number of compounds; $r_i$ is the ranking position of the $i^{th}$ active compound according to the scoring ranking.

Enrichment Factor (EF) is another metric to evaluate model performance, which calculates the fold increase in the proportion of active compounds among the top n% of predicted compounds compared to the proportion of active compounds in the entire dataset. A higher EF value indicates better performance of the model in predicting active compounds.

$$EF = \frac{\sum_{i=1}^{n} \delta_i}{\chi n} \quad \text{where } \delta_i = \begin{cases} 1, & r_i \leq \chi N \\ 0, & r_i > \chi N \end{cases}$$

$\chi$ is the fraction of the ordered list thatis considered and goes from 0 to 1.

## A.2 PROOF FOR FGW-CLIP

First, we establish a lemma for the loss 1 induced by the inverse optimal transport problem.

**Lemma 2** *The loss in Equation 1 can be derived from the following optimization problem:*

$$\min_{\theta} \quad KL(\hat{P}||P^{\theta})$$
$$\text{subject to} \quad P^{\theta} = \arg \min_{P \in U(a)} \left( \langle C^{\theta}, P \rangle - \tau H(P) \right) \tag{7}$$

*where $C^{\theta} \in R^{N \times N}, C^{\theta}(i,j) = c - s_{ij}(\theta)$ and $\hat{P}(i,j) = \frac{f_{ij}}{N}$, where $f_{ij}$ are label which equals to 1 when $x_i$ and $x_j$ are related else 0. $I_i$ denotes the index associated with $x_i$. $U(\mathbf{a}) = \{\Gamma \in R_+^{N \times N} | \Gamma \mathbf{1}_N = \mathbf{a}\}$. Here $\mathbf{a}$ denotes a vector whose elements are the sums of labels, specifically defined as $\mathbf{a}(i) = \sum_{j=1}^{N} f_{ij}$.*

We introduce the lagrangian of equation 2 as follows:

$$L(P, d) = (\langle C^{\theta}, P \rangle - \tau H(P) - \sum_{i=1}^{N} d_i (\sum_{j=1}^{N} (P_{ij} - a_i)) \tag{8}$$

The KKT conditions can be obtained as follows:

$$\frac{\partial L(P, d)}{\partial P_{ij}} = C_{ij}^{\theta} + \tau log P_{ij} - d_i = 0 \tag{9}$$

Given that $\sum_{j=1}^{N} P_{ij} = a_i$, we can derive the following expression:

$$P_{ij} = \frac{a_i e^{-C_{ij}^{\theta}/\tau}}{\sum_{j=1}^{N} e^{-C_{ij}^{\theta}/\tau}} \tag{10}$$

By solving the optimization problem 7 according to definition, we can obtain the following results:

$$L_{iot} = -\frac{1}{N}\sum_{i=1}^{N} a_i log(\frac{\sum_{j\in I_i} e^{-C_{ij}^{\theta}/\tau}}{\sum_{j=1}^{N} e^{-C_{ij}^{\theta}/\tau}}) + Constant \tag{11}$$

To simplify the problem, we disregard $a_i$ and constant in $L_{iot}$ in practical applications, resulting in the following expression:

$$L_{iot} = -\sum_{i=1}^{N} log(\frac{\sum_{j\in I_i} e^{-C_{ij}^{\theta}/\tau}}{\sum_{j=1}^{N} e^{-C_{ij}^{\theta}/\tau}}) \tag{12}$$

### A.2.1 Proof for Proposition 1

According to lemma 2, we can transform the original optimization problem into the following problem:

$$\min_{\theta,\psi_1,\psi_2} \Big\{ -(1-\alpha)\sum_{i=1}^{N}\sum_{j\in I_i} log(\frac{e^{-C_{ij}^{\theta}/\tau}}{\sum_{j=1}^{N} e^{-C_{ij}^{\theta}/\tau}}) + \alpha GW(\Gamma_d^{\psi_1}, \Gamma_d^{\psi_2}, \Gamma^{\theta})$$
$$-\lambda_1 \sum_{i=1}^{N}\sum_{j\in I_i} log(\frac{e^{-C_{ij}^{\psi_1}/\tau}}{\sum_{j=1}^{N} e^{-C_{ij}^{\psi_1}/\tau}}) - \lambda_2 \sum_{i=1}^{N}\sum_{j\in I_i} log(\frac{e^{-C_{ij}^{\psi_2}/\tau}}{\sum_{j=1}^{N} e^{-C_{ij}^{\psi_2}/\tau}}) - \lambda_{ce}CE(y_{\psi 1}, f_{\psi 1}(X_1)) \Big\} \tag{13}$$

For the GW term, we can simplify it according to the definition as follows:

$$GW(\Gamma_d^{\psi_1}, \Gamma_d^{\psi_2}, \Gamma^{\theta}) = (\Gamma_d^{\psi_1} \circ \Gamma_d^{\psi_1} a^{\psi_1})^{\top} a^{\psi_1} + (\Gamma_d^{\psi_2} \circ \Gamma_d^{\psi_2} a^{\psi_2})\top a^{\psi_2} - 2tr(\Gamma^{\theta\top}\Gamma_d^{\psi_1}\Gamma^{\theta}\Gamma_d^{\psi_2}) \tag{14}$$

which $\circ$ Hadamard product. Disregarding the constant terms, we can simplify the optimization objective as follows:

$$GW(\Gamma_d^{\psi_1}, \Gamma_d^{\psi_2}, \Gamma^{\theta}) = -2tr(\Gamma^{\theta\top}\Gamma_d^{\psi_1}\Gamma^{\theta}\Gamma_d^{\psi_2}) \tag{15}$$

Considering the symmetric positions of $i$ and $j$, a classic technique is to transform the original optimization problem into the following form:

$$\min_{\theta,\psi_1,\psi_2} \Big\{ -\frac{1-\alpha}{2}(\sum_{i=1}^{N}\sum_{j\in I_i} log(\frac{e^{-C_{ij}^{\theta}/\tau}}{\sum_{j=1}^{N} e^{-C_{ij}^{\theta}/\tau}}) + \sum_{j=1}^{N}\sum_{i\in J_j} log(\frac{e^{-C_{ij}^{\theta}/\tau}}{\sum_{i=1}^{N} e^{-C_{ij}^{\theta}/\tau}})) - 2\alpha tr(\Gamma^{\theta\top}\Gamma_d^{\psi_1}\Gamma^{\theta}\Gamma_d^{\psi_2})$$
$$-\lambda_1 \sum_{i=1}^{N}\sum_{j\in I_i} log(\frac{e^{-C_{ij}^{\psi_1}/\tau}}{\sum_{j=1}^{N} e^{-C_{ij}^{\psi_1}/\tau}}) - \lambda_2 \sum_{i=1}^{N}\sum_{j\in I_i} log(\frac{e^{-C_{ij}^{\psi_2}/\tau}}{\sum_{j=1}^{N} e^{-C_{ij}^{\psi_2}/\tau}}) - \lambda_{ce}CE(y_{\psi 1}, f_{\psi 1}(X_1)) \Big\} \tag{16}$$

The above expression is consistent with the form of the overall loss obtained by FGW-CLIP. Given that $i \in I_i, j \in J_j$ and $C^{\theta}(i,j) = c - x_{\psi_1,i}x_{\psi_2,j}^T$, by reorganizing equation 16, it can be observed that:

$$\min_{\theta,\psi_1,\psi_2} \Big\{ -\frac{1-\alpha}{2}(\sum_{i=1}^{N}\sum_{j\in I_i} (x_{\psi_1,i}x_{\psi_2,j}^T - c)/\tau + \sum_{j=1}^{N}\sum_{i\in J_j} (x_{\psi_2,i}x_{\psi_1,j}^T - c)/\tau) + \alpha GW(\Gamma_d^{\psi_1}, \Gamma_d^{\psi_2}, \Gamma^{\theta})$$
$$+\frac{1-\alpha}{2}(\sum_{i=1}^{N}\sum_{j\in I_i} log(\sum_{j=1}^{N} e^{-C_{ij}^{\theta}/\tau}) + \sum_{j=1}^{N}\sum_{i\in J_j} log(\sum_{i=1}^{N} e^{-C_{ij}^{\theta}/\tau}))$$
$$-\lambda_1 \sum_{i=1}^{N}\sum_{j\in I_i} log(\frac{e^{-C_{ij}^{\psi_1}/\tau}}{\sum_{j=1}^{N} e^{-C_{ij}^{\psi_1}/\tau}}) - \lambda_2 \sum_{i=1}^{N}\sum_{j\in I_i} log(\frac{e^{-C_{ij}^{\psi_2}/\tau}}{\sum_{j=1}^{N} e^{-C_{ij}^{\psi_2}/\tau}}) - \lambda_{ce}CE(y_{\psi 1}, f_{\psi 1}(X_1)) \Big\} \tag{17}$$

Since $x_{\psi_1}, x_{\psi_2}$ are $L_2$ normalized, disregarding constants, we can derive that:

$$
\min_{\theta,\psi_1,\psi_2} \Bigg\{ \frac{1-\alpha}{2} \Big( \sum_{i=1}^{N} \sum_{j\in I_i} |x_{\psi_1,i} - x_{\psi_2,j}|^2/\tau^2 + \sum_{j=1}^{N} \sum_{i\in J_j} |x_{\psi_2,i} - x_{\psi_1,j}|^2/\tau^2 \Big) + \alpha GW(\Gamma_d^{\psi_1}, \Gamma_d^{\psi_2}, \Gamma^\theta)
$$

$$
+ \frac{1-\alpha}{2} \Big( \sum_{i=1}^{N} \sum_{j\in I_i} log(\sum_{j=1}^{N} e^{-C_{ij}^\theta/\tau}) + \sum_{j=1}^{N} \sum_{i\in J_j} log(\sum_{i=1}^{N} e^{-C_{ij}^\theta/\tau}) \Big)
$$

$$
- \lambda_1 \sum_{i=1}^{N} \sum_{j\in I_i} log\Big( \frac{e^{-C_{ij}^{\psi_1}/\tau}}{\sum_{j=1}^{N} e^{-C_{ij}^{\psi_1}/\tau}} \Big) - \lambda_2 \sum_{i=1}^{N} \sum_{j\in I_i} log\Big( \frac{e^{-C_{ij}^{\psi_2}/\tau}}{\sum_{j=1}^{N} e^{-C_{ij}^{\psi_2}/\tau}} \Big) - \lambda_{ce} CE(y_{\psi 1}, f_{\psi 1}(X_1)) \Bigg\}
$$

$$(18)$$

From the equation, we can deduce that $L_F GW$ is the optimization of a specific fused Gromov-Wasserstein distance under regularization conditions.

### A.3 EXPERIMENT DETAILS

For the training of FGW-CLIP, we use Adam optimizer at a learning rate of 0.001. The batch size is 32, and the training is conducted on 4 NVIDIA GeForce RTX 4090 24G GPUs. For the reaction part, the molecular encoder parameters are the same as those of Uni-Mol. The readout function is sum. For the enzyme part, the encoder used is ESM2, which is frozen during the training process. We added a linear layer after the embedding output by ESM2 to help with mapping. The training epoch is 100, and the last checkpoint is selected.

### A.4 LIMITATION

Currently, the framework scenario is only limited to the case of enzyme screening. In the future, it will be applied to more extensive enzyme prediction and design tasks.

