# OpenReview forum: "FGW-CLIP: Enhancing Enzyme Screening via Fused Gromov-Wasserstein Contrastive Learning"
_ICLR.cc/2025/Conference — ICLR 2025 Conference Withdrawn Submission_

### Official Review · Reviewer_F1cX · 2024-10-27

**Soundness:** 2
**Presentation:** 1
**Contribution:** 2
**Rating:** 3
**Confidence:** 4

**Summary:**

### Summary:
Leveraging CLIPZyme, this paper introduces a new CLIP method that combines Fused Gromov Wasserstein Distance and structure alignments for enzyme screening. FGW-CLIP incorporates alignments, including alignments between reactions and enzymes, as
well as internal alignment of enzyme-reaction representations.

**Strengths:**

### Advantage:
1. Better results than baselines.
2. Negative sample construction is novel, using EC numbers which can leverage hierarchical information in protein family.

**Weaknesses:**

### Disadvantage:
1. Paper writing needs to be improved.
2. Sec 3.2 is too long for model description.
3. No proper citation for InfoNCE [3].
4. Please avoid to use the word 'alignment' if you are not using MSA, it is not appropriate. You are just enforcing similarities, not actually doing alignments.
5. Sec 3.3 is too redundant, most are not new.
6. The paper is an unfinished work to me.

**Questions:**

### Question:
(1) Given the retrieval property of this task, you should try to used retrieval-based metrics (in [2]) to rank enzymes given the reaction (or vice versa). You should also try on CLIPZyme, see if FGW-CLIP can outperforms CLIPZyme on retrieval metrics.


### You should cite the following papers of using contrastive learning for screening published at Neurips:
[1] Yang, Jason, Ariane Mora, Shengchao Liu, Bruce J. Wittmann, Anima Anandkumar, Frances H. Arnold, and Yisong Yue. "CARE: a Benchmark Suite for the Classification and Retrieval of Enzymes." arXiv preprint arXiv:2406.15669 (2024).

[2] Hua, Chenqing, Bozitao Zhong, Sitao Luan, Liang Hong, Guy Wolf, Doina Precup, and Shuangjia Zheng. "Reactzyme: A benchmark for enzyme-reaction prediction." arXiv preprint arXiv:2408.13659 (2024).

[3] Oord, Aaron van den, Yazhe Li, and Oriol Vinyals. "Representation learning with contrastive predictive coding." arXiv preprint arXiv:1807.03748 (2018).

---

### Official Review · Reviewer_wDC5 · 2024-10-29

**Soundness:** 3
**Presentation:** 2
**Contribution:** 3
**Rating:** 5
**Confidence:** 4

**Summary:**

This paper introduces FGW-CLIP, a novel framework for enzyme-reaction contrastive learning. The framework uses intra-contrastive learning loss within enzymes and reactions, as well as inter-contrastive learning between enzymes and reactions. Additionally, FGW-CLIP aligns the overall learning objectives with a fused Gromov-Wasserstein distance optimization problem under specific constraints, enhancing the generalization capability of the contrastive learning approach. The proposed method achieves superior performance on the EnzymeMap dataset compared to the previous SOTA method, CLIPZyme.

**Strengths:**

1. The multiple contrastive learning objectives introduced in FGW-CLIP are reasonable and effective.

2. The introduction of optimizing the Gromov-Wasserstein distance is novel, and from the ablation study, it is the most effective and important part.

**Weaknesses:**

1. The figures and corresponding captions in this paper should be improved. Especially the t-sne visualization in figure 2. It is better to have a more clear and professional version. More analysis of the learned representations can also be added.

2. More experiments should be added. it seems that some experiment settings and datasets used in  CLIPZyme are not tested in FGW-CLIP.

**Questions:**

1. The baseline model CLIPZyme uses a DMPNN for reaction representations, while FGW-CLIP uses a pre-trained encoder Uni-Mol to get the representation. It would be interesting to compare FGW-CLIP and CLIPZyme with exact same encoders.

---

### Official Review · Reviewer_m4cm · 2024-11-03

**Soundness:** 3
**Presentation:** 3
**Contribution:** 1
**Rating:** 3
**Confidence:** 5

**Summary:**

FGW-CLIP takes the model CLIPZyme and extends it by modifying the reaction encoder and enzyme encoder as well as including three additional losses -- the FGW loss, an EC loss and an internal loss. They show improved performance on the EnzymeMap dataset on the task of virtual enzyme screening.

**Strengths:**

There are a number of insights in the paper which are important and have an impact on the task:
1. The authors identified that reactions may be similar if they are recognized by the same enzyme or fall under the same EC number -- the authors add an internal loss to address this which pushes representations of similar reactions and enzymes to be closer. This has a positive effect on the measured metrics.
2. The authors note that adding an auxiliary EC loss improves performance -- this makes sense and indeed it does.
3. The authors apply a novel FGW loss that aims to minimize the Gromov-Wasserstein distance between the enzyme and reaction spaces -- this is a novel method that has not before been applied to these problems (to the best of my knowledge).
4. The authors achieve improved results on all presented metrics.

**Weaknesses:**

There are a number of weaknesses in the paper:
1. The evaluation setup compares FGW-CLIP to CLIPZyme, yet since the publication of CLIPZyme two new benchmarking papers have been released that provide a comprehensive evaluation setup on this task: Yang et al (CARE: a Benchmark Suite for the Classification and
Retrieval of Enzymes) and Hua et al (ReactZyme: A Benchmark for Enzyme-Reaction Prediction). CREEP (in Yang et al) also includes auxiliary tasks as a part of the model and shows improvement. Please provide results comparing your model to more recent models on this task.
2. The paper excludes additional benchmarks provided in the original CLIPZyme paper, such as performance on sequence and structure similarity splits of the screening set, unannotated EnzymeMap and Terpene Synthases.
3. Ablation study -- it is difficult to interpret the ablation study since components of the loss do not seem to be additive to the performance. For example why is BEDROC-85 R_E + R_R better than all other methods except for FGW-CLIP? This seems strange. Additionally, how much of the performance gain is attributed to using different encoders compared to a different loss? How does the CLIPZyme architecture perform with the FGW loss?
4. Please provide additional analysis to explain the performance difference and its significance. The BEDROC values represent an experimental screening setup of 10,000 proteins which is not a realistic budget -- how does FGW-CLIP do on 1,000 proteins?
5. Please provide tsne plots that show groups that the model was not trained on -- it is not completely surprising that the representations of FGW-CLIP cluster by EC since it was trained to do that while CLIPZyme was not. What do the representations look like if we cluster by sequence similarity, structure similarity or reaction similarity? What can we learn from these plots?

**Questions:**

The following questions (as well as the above questions) should be thoroughly answered to prove that the improvement presented in the paper is meaningful:
1. Perhaps provide qualitative results for a number of different reactions (across different EC classes for example) showing what screening budget would be needed with FGW-CLIP vs CLIPZyme
2. What impact does changing the encoders have as compared to the losses? Compare your losses with different architectures to show that the loss is what is driving the performance.
3. Why do you think that the 3D unimol representations are better than alternative representations? Since presumably the 3D conformers are random (since they do not exist in EnzymeMap) -- the 3D information is largely irrelevant? Could it be that there is leakage between the training molecules and the molecules that UniMol is trained on?

---

### Official Review · Reviewer_x35f · 2024-11-04

**Soundness:** 3
**Presentation:** 2
**Contribution:** 2
**Rating:** 3
**Confidence:** 3

**Summary:**

The paper presents an enhanced contrastive learning framework for enzyme screening, distinguishing itself from prior work [1] through two significant advancements: 1. The author adopted the pre-trained, transformer-based Uni-Mol framework to generate the representations of the 3D molecules. 2. The author proposed a fused loss function L_{FGW} that integrates multiple alignments, including reaction-enzyme alignment (Eq. 1), internal alignments for both enzyme and reaction (Eq.2), as well as regularization terms for minimizing the Gromov-Wasserstein distance between embedding spaces.

In the experimental results, Table 3 highlights the method '{R-E},' which shows a marginal improvement in the Boltzmann-Enhanced Discrimination of ROC scores when employing the 3D-based Uni-Mol framework as opposed to the 2D-based directed message-passing neural network (DMPNN) for reaction encoding. Table 1 demonstrates the overall improvement in performance in adopting the FGW-CLIP framework.

**Strengths:**

This paper's major contribution is its use of Uni-Mol, which incorporates 3D structural information into the contrastive learning framework.

The paper conducted an ablation study, detailed in section 4.2, which demonstrates the contribution of different components and loss terms of the overall loss function L{FGW}.

**Weaknesses:**

The novelty of this paper appears limited. It conducts minor modifications on pre-existing frameworks [1]. Regardless of the novelty, one significant drawback/question of this paper is: in [1], table 1: ‘Enzyme virtual screening performance compared to using EC prediction alone and together with CLIPZyme’ reports that the BEDROC85 score can increase from 44.69 to 57.03, and the BEDROC20 increase from 62.98 to 78.50, even with basic information EC level information, which I believe is the major contribution of the previous framework. This paper, however, fails to discuss and conduct experiments on this part. In addition, in section 3.3.2, equation 4 incorporates multi-level classification loss, L_{EC}, as one of the fused loss L_{FGW} contributions. This indicates that the ground truth of EC level is known. I am wondering whether the author can further clarify this part.

The authors state that the experimental settings were aligned with those reported in [1]. However, the number of protein sequences used in the Enzyme Screening set appears to differ from those reported in [1]. Could the authors clarify if this difference is due to a typographical error or if adjustments were made to the dataset? In addition, it is better to include the training details in optimizing L_{FGW}. E.g., which optimizer is used, how it is set up, and how the regularization term \lambda is chosen.

It is better to provide a detailed description of Figure 2. E.g., include a legend of the figure.

---

[1] Mikhael, Peter G., Itamar Chinn, and Regina Barzilay. "CLIPZyme: Reaction-Conditioned Virtual Screening of Enzymes." arXiv preprint arXiv:2402.06748 (2024).

**Questions:**

My questions are listed in the weakness part.

---

### Note · Authors · 2024-11-21

I have read and agree with the venue's withdrawal policy on behalf of myself and my co-authors.